# Automatic Detection and Classification of Dead Nematode-Infested Pine Wood in Stages Based on YOLO v4 and GoogLeNet

Xianhao Zhu [1,2], Ruirui Wang [1,2,*], Wei Shi [3], Qiang Yu [1,2], Xiuting Li [1,2] and Xingwang Chen [1,2]

1   College of Forestry, Beijing Forestry University, Beijing 100083, China; zxh2020@bjfu.edu.cn (X.Z.);
    yuqiang@bjfu.edu.cn (Q.Y.); lixiuting97@bjfu.edu.cn (X.L.); chenxingwang@bjfu.edu.cn (X.C.)
2   Beijing Key Laboratory of Precision Forestry, Beijing Forestry University, Beijing 100083, China
3   Beijing Ocean Forestry Technology Co., Ltd., Beijing 100083, China; bnuvictor@hotmail.com
*   Correspondence: ruiwang@bjfu.edu.cn

**Abstract:** Pine wood nematode disease has harmed forests in several countries, and can be reduced by locating and clearing infested pine trees from forests. The target detection model of deep learning was utilized to monitor a pine nematode-infested wood. The detecting effect was good, but limited by low-resolution photos with poor accuracy and speed. Our work presents a staged detection and classification approach for a dead nematode-infested pine wood based using You Only Look Once version 4 (YOLO v4) and Google Inception version 1 Net (GoogLeNet), employing high-resolution images acquired by helicopter. Experiments showed that the detection accuracy of the staged detection and classification method and the method using only the YOLO v4 model were comparable for a dead nematode-infested pine wood when the amount of data was sufficient, but when the amount of data was limited the detection accuracy of the former was higher than that of the latter. The staged detection and classification method retained the fast training and detection speed of the one-stage target detection model, further improving the detection accuracy with limited data volume, and was more flexible in achieving accurate classification, meeting the needs of forest areas for pine nematode disease epidemic prevention and control.

**Keywords:** dead nematode-infested pine wood; deep learning; target detection; recognition classification

## 1. Introduction

Pine wood nematode disease is characterized as pine tree "cancer." Pine trees might die about 40 days following infection with this disease [1], which has occurred in many countries including the United States, Japan, and Korea, and has caused substantial damage to forest resources and the ecological environment in each country [2]. The intermediate host of pine wood nematode (*Bursaphelenchus xylophilus*) transmission in forest regions is usually the pine brown aspen, which is particularly destructive and difficult to control [3]. Nematode-infested healthy pine trees die within three months, and the disease spreads to surrounding pine trees if the infested trees are not removed from the forest in time [4]. Therefore, effective diagnosis of pine nematode-infested wood, and timely and thorough cleanup of dead nematode-infested pine wood are crucial for controlling the disease [5].

Initial monitoring of pine nematode-infested wood mainly relies on visual observation and regular inspection by forest rangers, which are inefficient and do not ensure accuracy of judgment [6]. With the development of conventional machine learning methods and aerial photography technology, scientific research work on monitoring pine nematode-infested wood has made some progress, including support vector machines [7–9], random forests [10–12], object-oriented segmentation [13,14], clustering algorithms [15–17], and others. However, typical machine learning approaches only analyze the differences between

the low-level properties of diseased and healthy pine wood, such as color and texture for diagnosing diseased wood, which has some bearing on detection accuracy [18].

With the advent of deep learning, the issues of computational duplication and inefficiency of simple low-level feature examination generally seen in classical machine learning have been successfully addressed. Compared with typical machine learning techniques, deep learning is regarded as a very accurate recognition method [19], and it is often employed in pine nematode-infested wood monitoring. Deep learning includes convolutional neural networks [20–22], spatial-context-attention networks [18], semantic segmentation networks [23–25], target detection models [26–29], among others. Target detection models are widely used and effective in the field of disease identification [30], but their detection effect is limited by low-resolution images and cannot reconcile detection accuracy and speed [31].

Currently, deep learning-based target detection models mainly consist of one-stage detection or two-stage detection models. Two-stage-based target detection models such as Faster Region-Based-CNN (Faster R-CNN) [32] are characterized by first forming target candidate frames and then performing target classification. These models have higher overall recognition detection accuracy at the expense of increased computational cost, longer training time, slower detection, and more demanding hardware requirements. One-stage-based target detection models such as YOLO v4 [33] and Single Shot MultiBox Detector (SSD) [34] enable simultaneous target border prediction and classification. Their training and detection are relatively faster, their hardware requirements are less demanding, and they are easier to deploy in the experimental environment, but they are not as accurate as two-stage models.

In this study, to address the problem that the target detection model cannot balance speed and detection accuracy, a staged detection and classification method is proposed to combine the YOLO v4 model with the GoogLeNet model for target detection and wilting degree classification of dead nematode-infested pine wood, with high spatial resolution images acquired by helicopters as the data base. The main objectives of this study are to:

1.  compare the performance of two target detection models, YOLO v4 and SSD, in the task of dead nematode-infested pine wood detection;
2.  compare the performance of five recognition classification models, GoogLeNet and ShuffleNet, ResNet50, MobileNet-v2, ResNet18, in the task of wilting degree classification of dead nematode-infested pine wood;
3.  investigate whether the staged detection and classification approach can further improve the detection accuracy of the target detection task while retaining the fast training speed and detection speed of the one-stage target detection model.

The rest of the paper is organized as follows. The experimental materials and methods are described in Section 2, the experimental results are in Section 3, and the discussion and conclusions are in Sections 4 and 5, respectively.

## 2. Materials and Methods

### 2.1. Data Collection

#### 2.1.1. Overview of the Study Region

The experimental site was situated in the Baihe Conservation Management Station in the northern part of the Changbai Mountain National Nature Reserve in the Chinese province of Jilin. This area has a temperate monsoon climate zone in the core of the reserve, which conducive to the propagation of pine wood nematode outbreaks. The total area under the control of Baihe Conservation Management Station is 14,487 ha. The core region is 4762 ha and the experimental area is 9725 ha. The coordinates for the study area are $42°10'41''$ $42°11'29''$ N and $128°0'19''$ $128°3'34''$ E. The geographical location of the study area and true color photographs from a helicopter are displayed in Figure 1.

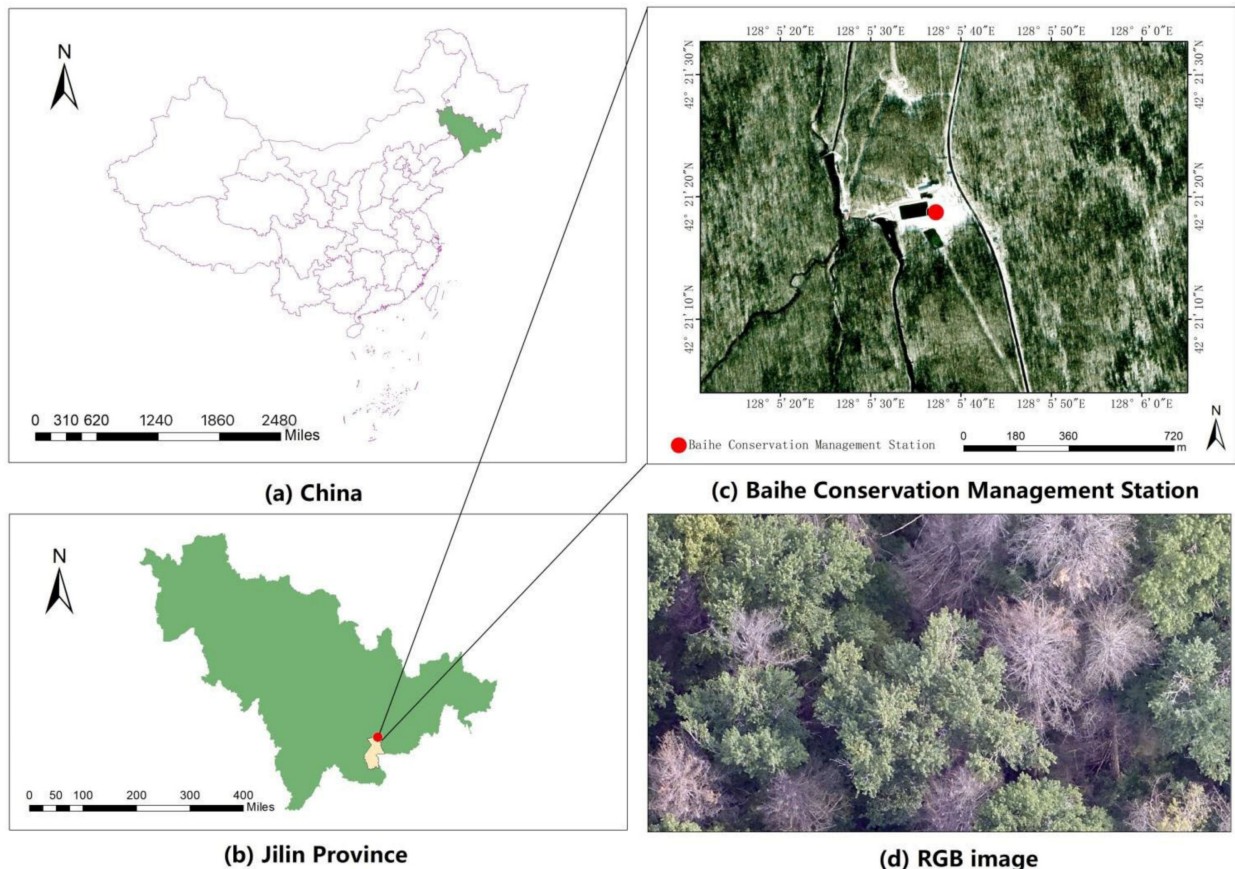

**Figure 1.** Geographical location and image map of the study area. (**a**) Map of China with Jilin Province in green color; (**b**) map of Jilin Province with the Changbai Mountains in beige color and the study area in red color; (**c**) geographical location of the Baihe Conservation Management Station; (**d**) RGB image of the study area obtained by helicopter.

### 2.1.2. Data Acquisition

A Bell 206 L4 helicopter was used as the aerial flight platform for data collection in the study area. The weather was clear and without clouds at the time of aerial photography, with ample light and no wind, which was excellent for helicopter aerial photography activities. A total of three flight strips were laid in the flight area. The flight parameters were then selected. According to the requirements of the project, the flight parameters were developed by combining the aerial digital camera system and the aircraft's performance indicators while ensuring that the ground resolution of the lowest point in the survey area and the heading and parallax overlap of the highest point met the technical requirements. This resulted in the flight height being at 400 m, the parallax overlap being about 45%, and the heading overlap being 65%.

The helicopter was equipped with a Feith camera with 100 million pixels and 3 bands of red, green, and blue. It captured 2956 images of 11,608 × 8708 in size. The center wavelengths of the red, green, and blue bands were 660 nm, 550 nm, and 440 nm, respectively. The resolution of the obtained airborne high-resolution image was 0.03 m, containing three bands of red, green, and blue, and the total size of the study area was 11.8 km$^2$. The size of the image data was 37.4 GB.

### 2.1.3. Dataset Production

In this study, multiple JPC images containing dead nematode-infested pine wood were derived using the high-resolution images consisting of red, green, and blue bands taken from a helicopter. A target detection model dataset and a recognition classification model dataset were constructed with the exported JPG images. The former contained a total of

884 JPG photos of 640 × 640 in size. The ratio of training set, validation set, and test set was 8:1:1, and the number of images in each set was 707, 89, and 88, respectively. The split training set was expanded to 2828 images after enhancement operations of flipping, scaling and color dithering, while the validation and test sets were not enhanced to obtain unbiased estimates. The augmented set contained 3005 pictures and 10,145 dead nematode-infested pine wood tags. A schematic representation of the sample labeling method for the target detection model is shown in Figure 2.

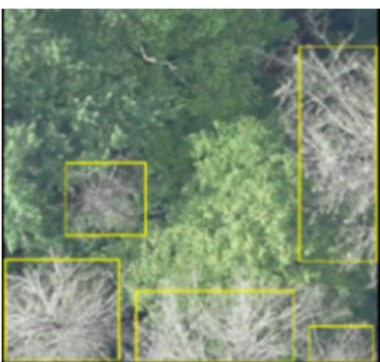

**Figure 2.** Sample annotation schematic of the target detection model dataset.

The recognition classification model dataset contained a total of 400 JPG images 224 × 224 in size. The ratio of the training and validation sets was 3:1, and the ratio of dry and recently dead nematode-infested pine wood was 1:1, as indicated in Table 1.

**Table 1.** Recognition classification model dataset.

| Category | Example Images | Number of the Training Set Samples | Number of the Validation Set Samples |
|---|---|---|---|
| Dry nematode-infested pine wood | | 140 | 60 |
| Recently dead nematode-infested pine wood | | 140 | 60 |

### 2.2. Methods

#### 2.2.1. Experiment Content

We addressed the problem that the target detection model cannot combine speed and detection accuracy, and used a staged detection and classification method combining YOLO v4 model and GoogLeNet model for target detection and the wilting degree classification of dead nematode-infested pine wood using high spatial resolution images acquired by helicopter as the data base. The technical roadmap is shown in Figure 3.

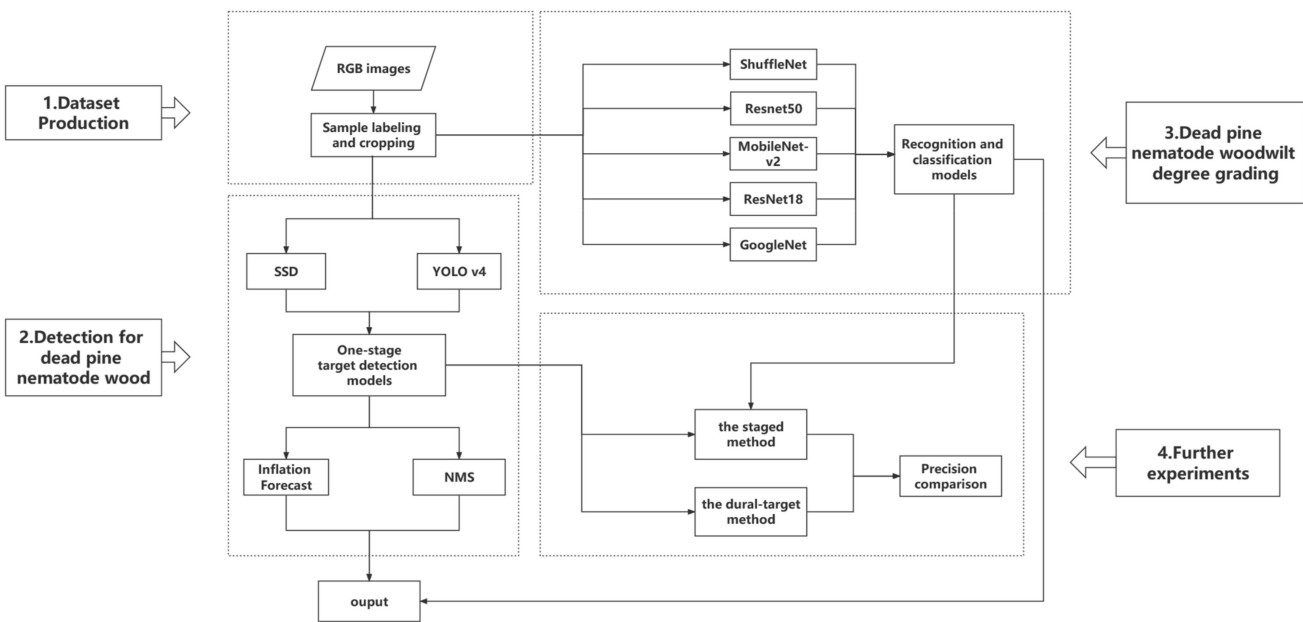

**Figure 3.** Technology roadmap.

Dataset production. Using the high-resolution images acquired from the helicopter, the target detection model dataset and recognition classification model dataset were produced, includinga training set, validation set, and test set.

1.  Target detection of dead nematode-infested pine wood. YOLO v4 was trained with the produced target detection model dataset. The trained YOLO v4 model was then used to detect dead nematode-infested pine wood in forest images and compared with the SSD target detection model.
2.  Wilting degree classification of dead nematode-infested pine wood. A GoogLeNet model was trained using the produced recognition classification model dataset, and then used to classify the detected dead nematode-infested pine wood as dry or recently dead to complete wilting degree classification. It was then compared with four other recognition classification models, namely ShuffleNet, ResNet50, MobileNet-v2, and ResNet18.
3.  Further experiments. Under the same conditions, the detection and classification experiments of dead nematode-infested pine wood were conducted using the staged detection and classification method and the dual target detection and classification method using the YOLO v4 model only. The experimental results were compared to investigate whether the staged detection classification method could improve detection accuracy while retaining the fast training speed and detection speed of the one-stage target detection model.

2.2.2. Target Detection Model

Introduction to YOLO v4

The YOLO v4 algorithm is built on YOLO v3 and has been further improved by merging the greatest algorithmic ideas of recent years. The mean average precision (mAP) was increased to 44% on the MS COCO target detection dataset without lowering the frame rate, i.e., the number of frames per second (FPS). The key components of the YOLO v4 network are: (1) the backbone feature extraction network Cross Stage Partial DarkNet53 (CSPDarkNet53), (2) the enhanced feature extraction network Spatial Pyramid Pooling (SPP) and Path Aggregation Network (PANet), and (3) the prediction network YOLO head. The specific network model structure of YOLO v4 is illustrated in Figure 4.

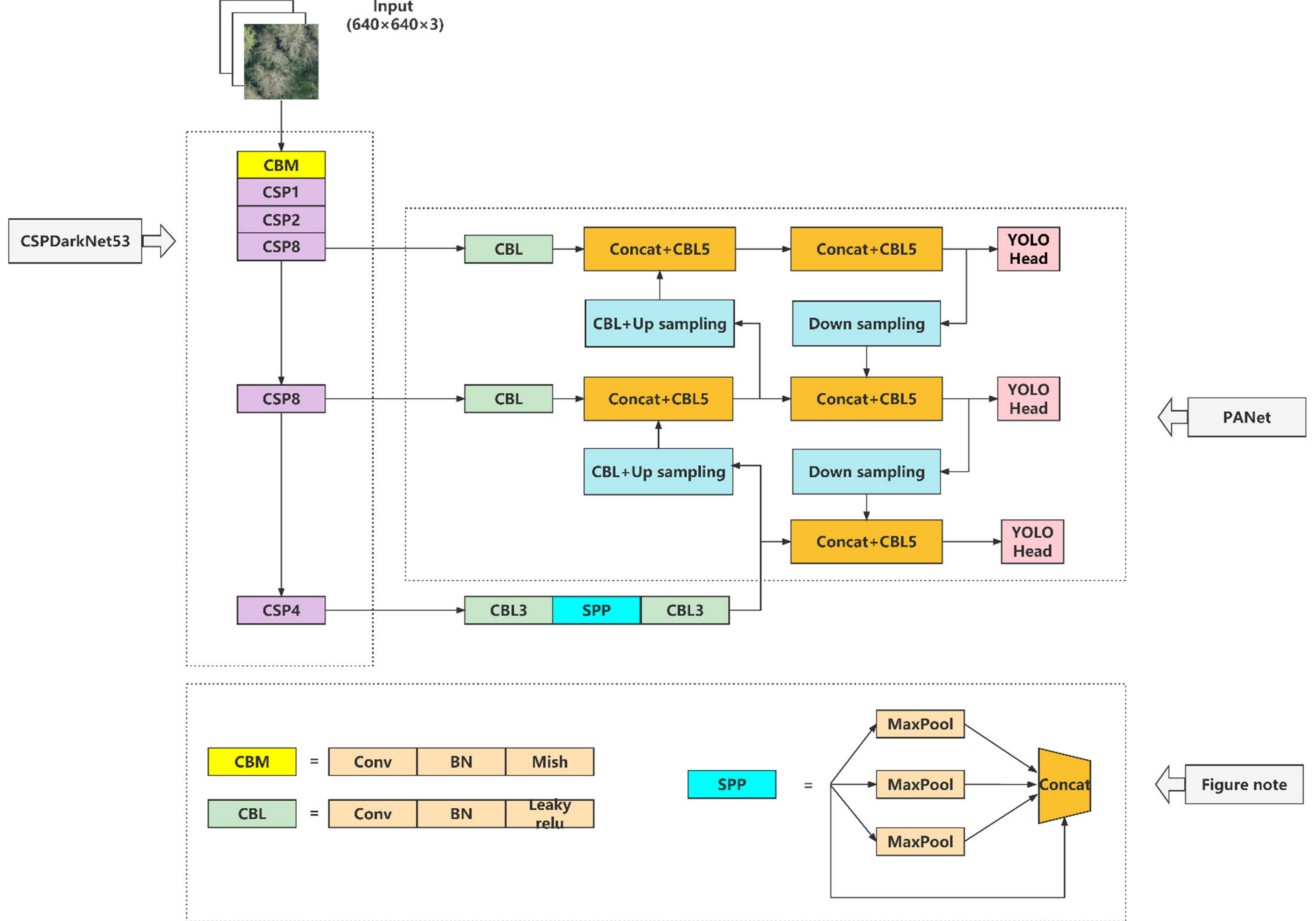

**Figure 4.** YOLO v4 model structure diagram.

The backbone network of YOLO v4 is CSPDarkNet53 [35], which is used to initially extract image features from the input. YOLO v4 adopts the CSP structure to change the residual structure: the input image features are convolved and divided into two branches, the upper branch is stacked with the residual component, and the lower branch is convolved and then fused with the higher branch for features. There are two advantages to choosing the CSPDarkNet53 network structure as the backbone. First, it improves the ability of the convolutional network to extract features without compromising detection accuracy, which speeds up the detection. Second, it decreases the computational cost of the overall model, which makes the model simpler for hardware setup.

The SPP [36] structure is a feature re-extraction of the last feature layer previously extracted from the CSPDarkNet53 structure. After three feature layer convolutions, four scales are used to accomplish the maximum pooling operation, and the pooling kernel sizes are $1 \times 1$, $5 \times 5$, $9 \times 9$, and $13 \times 13$, respectively, to maximize the perceptual field and isolate the more visible contextual features.

The PANet [37] structure is a cyclic pyramid structure composed of convolution, upsampling, feature layer fusion, and downsampling operations that iteratively extract feature images and finally output three feature layers that are fed into the YOLO Head to evaluate each feature layer's prediction frame and output the target's final prediction frame.

Target Detection Optimization Strategy

The target image is relatively large, and we used the sliding window approach to detect targets. However, if sliding window recognition is applied directly on the target image, it affects the recognition of discolored nematode-infested standing pine wood trees at the edges due to less contextual information in the edge region of each image block

obtained through cropping. Therefore, when target detection was performed on the image, the expansion prediction method was used for inflation prediction. The sliding window size was set to 640 × 640, and the sliding step size set to 320 each time. Only the recognition result of the central part of 320 × 320 size was retained for each recognition, and the discarded area became the central area of other sliding windows. This avoids the problem of inaccurate recognition results of discolored standing pine wood trees due to the feature extraction problem at the boundary. A more graphic diagram of inflation prediction is shown in Figure 5.

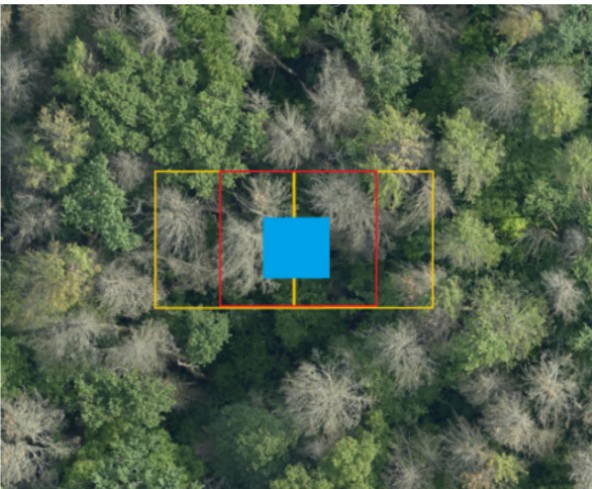

**Figure 5.** Schematic diagram of inflation prediction.

In the target detection process, executing a detection task resulst in a significant number of redundant detectors that need to be deleted by the Non-Maximum Suppression (NMS) algorithm. To achieve the suppression effect, each list recursively selects the top scoring checkboxes and removes any that overlap by more than the threshold. The comparison of the effect before and after NMS optimization is shown in Figure 6.

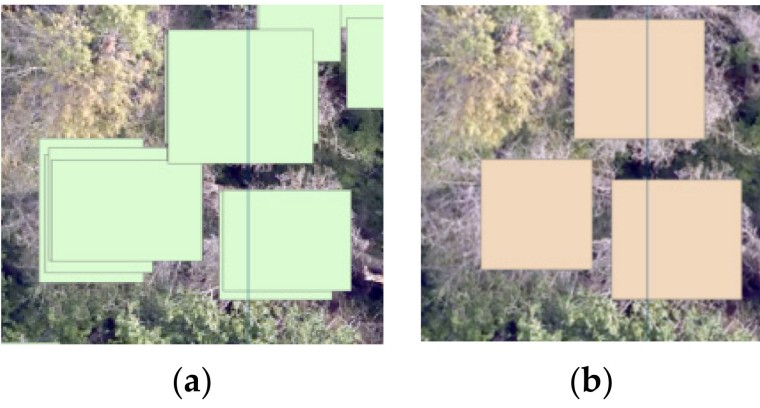

(**a**) (**b**)

**Figure 6.** NMS optimization effect comparison chart. (**a**) Before NMS (**b**) After NMS.

### 2.2.3. Introduction to the Recognition Classification Model GoogleNet

The main structure of GoogLeNet model used in this paper includes an input layer, five sets of convolutional modules, and an output layer with 22 parametric layers and five pooling layers [38]. The input layer is a 224 × 224 × 3 image. The group 1 and group 2 convolutional modules include convolutional and maximum pooling layers, the group 3, group 4, and group 5 convolutional modules mainly consist of Inception v1 module structure and maximum pooling layer, and the output layer consists of average pooling, drop-out, and fully connected layers. The network structure diagram of GoogLeNet is illustrated in Figure 7.

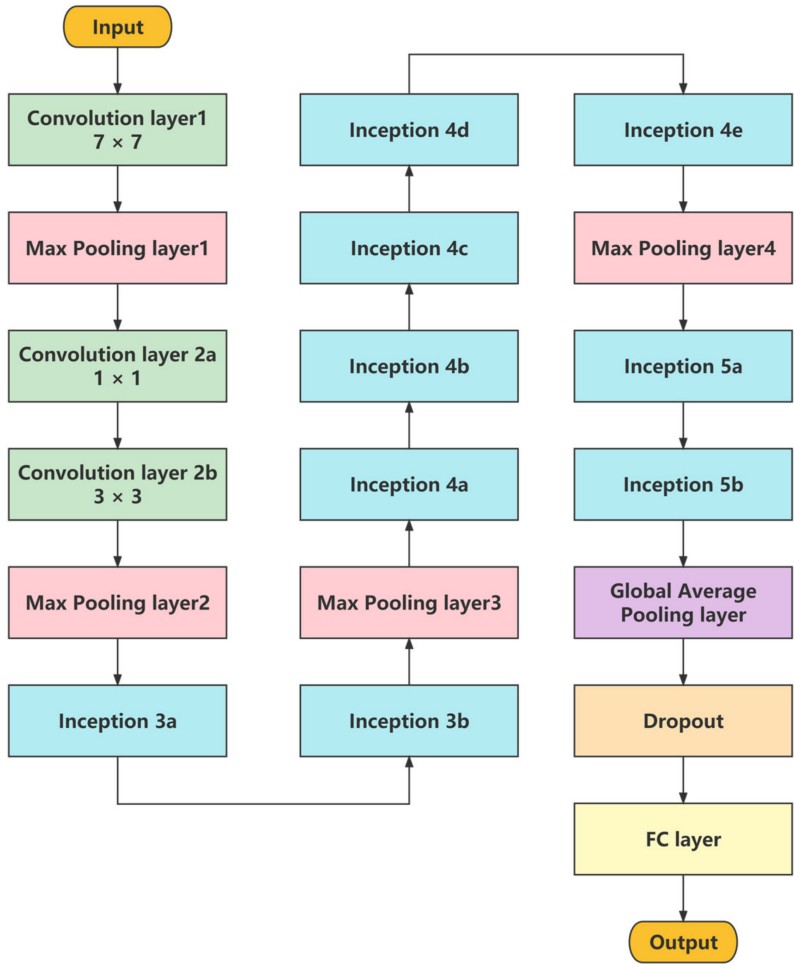

**Figure 7.** GoogLeNet model structure diagram.

The Inception v1 module [38] consists of $1 \times 1$, $3 \times 3$, and $5 \times 5$ multi-scale convolutional kernels and $3 \times 3$ pooling layers, which enable the characteristics of the target object to be captured by the appropriate convolutional kernel during the image recognition process, no matter how large the target object is. In addition, the parallel operation of pooling layer and convolutional layer may extend the channels and can assure the operation's efficiency at the same time. The Inception v1 module's structure is depicted in Figure 8.

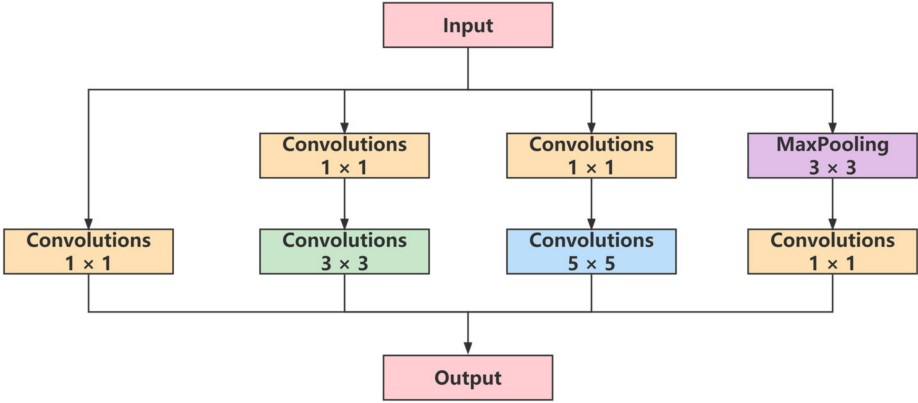

**Figure 8.** Inception v1 structure diagram.

2.2.4. Test Environment and Parameter Settings

The CPU was an AMD Ryzen 7 5800H with Radeon Graphics, the GPU is NVIDIA GeForce RTX 3060 Laptop GPU with 6 GB memory, the main board was a LENOVO, and the memory was 16 GB. AMD's headquarters and NVIDIA's headquarters are both located in Santa Clara, California, USA. Lenovo is headquartered in Beijing District, China. In deep learning training, hyperparameter selection is challenging and time-consuming because the optimal combination of hyperparameters depends not only on the model itself, but also on the data and hardware environments. Several experiments were conducted to determine the model's hyperparameters for this study.

Target detection models were trained using a variable learning rate controlled by equal interval decay. The initial learning rate was set to 0.0001 with an adjustment interval of 30 rounds and a decay factor of 0.8, i.e., after every 30 rounds, the learning rate decayed to 80% of its current value. The optimizer employed stochastic gradient descent with momentum (SGDM) with a momentum of 0.9. The weight decay value was 0.0005, the batch size was set to 4, and the number of epochs was set to 200.

The recognition classification model training adopted a fixed learning rate mode and randomly splits the training and validation sets into several batches of size 16. The number of rounds (or epochs) was set to 30, the learning rate was set to 0.001, and the optimizer adopted SGDM.

2.2.5. Accuracy Inspection

The test images were manually labeled with dead nematode-infested pine wood, and the target detection results were compared with manually-labeled images to determine the accuracy of the detection and validate the results. The precision rate, recall rate, given in (1)–(2), respectively, were utilized to evaluate the model recognition results. The precision rate was used to measure the accuracy of the model to detect dead nematode-infested pine wood, i.e., the detection accuracy. The recall rate was used to evaluate the comprehensiveness of the model in detecting dead nematode-infested pine wood, i.e., the overall detection rate.

$$\text{Precision} = \frac{\text{TPA}}{\text{TPA} + \text{FPA}} \tag{1}$$

$$\text{Recall} = \frac{\text{TPA}}{\text{TPA} + \text{FNA}} \tag{2}$$

where TPA, FPA, and FNA have specific meanings shown in Table 2.

**Table 2.** The evaluation indicator table of target detection models.

| Evaluation Indicators | Indicator Description |
|---|---|
| TPA | Dead nematode-infested pine wood is correctly identified as dead nematode-infested pine wood |
| FPA | Other features are incorrectly identified as dead nematode-infested pine wood |
| FNA | Dead nematode-infested pine wood is misidentified as other features |

F1 scores are widely used as metrics in statistics to measure the accuracy of binary classification models. [14,21,22,27]. This also takes into account the precision rate and recall rate of the target detection model and refers to the harmonic mean of the precision rate and recall rate, as shown in Formula (3). In this study, the F1 score was used as an evaluation indicator to evaluate the detection effect of the target detection model, also known as the detection accuracy, which was statistically significant.

$$\text{F1} = \frac{2 \times \text{Precision} \times \text{Recall}}{\text{Precision} + \text{Recall}} \tag{3}$$

To evaluate the accuracy of wilting degree classification for dead nematode-infested pine wood, recognition classification results were compared with manual recognition results for verification. The model recognition results were evaluated using the classification accuracy measure defined as:

$$\text{Classification Accuracy} = \frac{\text{TPB} + \text{TNB}}{\text{TPB} + \text{FNB} + \text{TNB} + \text{FPB}} \tag{4}$$

where TPB, FNB, TNB, and FPB have the specific meanings shown in Table 3.

**Table 3.** Evaluation indicator table of recognition and classification models.

| Evaluation Indicators | | Indicator Description |
|---|---|---|
| | TPB | Dry nematode-infested pine wood is correctly identified as dry nematode-infested pine wood |
| | FNB | Dry nematode-infested pine wood is incorrectly identified as recently dead nematode-infested pine wood |
| | TNB | Recently dead nematode-infested pine wood is correctly identified as recently dead nematode-infested pine wood |
| | FPB | Recently dead nematode-infested pine wood is incorrectly identified as dry nematode-infested pine wood |

## 3. Results

In the training phase of the deep learning model, loss function and accuracy are essential indicators to evaluate the accuracy and effectiveness of the model, where the loss function is often used to estimate how much the model's predictions differ from reality. The loss function has an absolute value, and the smaller that value, the better the model fits the data.

### 3.1. Target Detection Model Performance Analysis

A performance analysis of the target detection model is shown in Figure 9. As the number of iterations increased, the loss function in both YOLO v4 and SSD gradually decreased. YOLO v4 outperformed SSD in the final trained model's detection accuracy, but SSD converged faster.

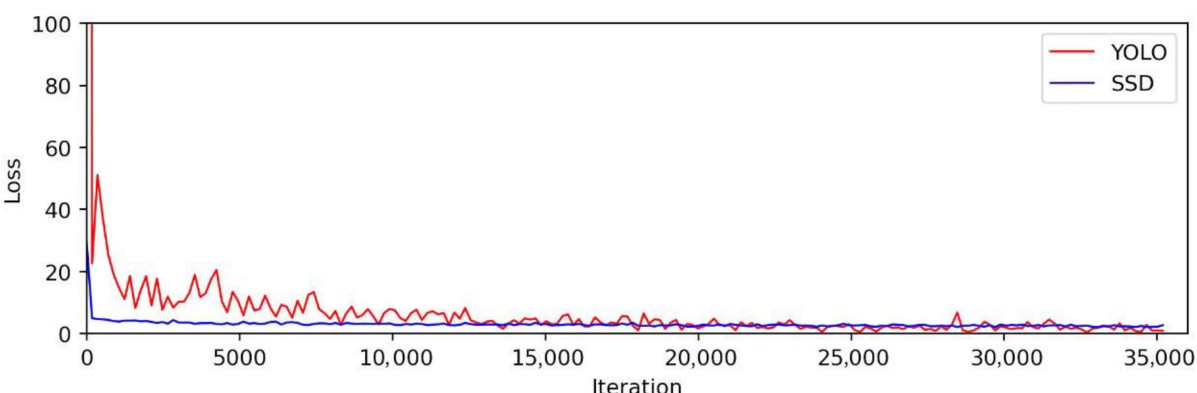

**Figure 9.** Performance comparison of object detection.

The training process of the target detection model is extremely time-consuming. Therefore, the training time is also an important metric to evaluate the model's performance. It can be seen in Table 4 that the average training time per round and the average time required per iteration for YOLO v4 was less than that of SSD when setting the same batch size, and YOLO v4 required less algorithm power than SSD with the same configuration and the same amount of data. This indicates the benefits of YOLO v4 in terms of training

speed. Additionally, for the same test image, the detection time of YOLO v4 was 229 s while the detection time of SSD was 280 s, which again demonstrates the advantage of YOLO v4 in terms of detection speed.

**Table 4.** Comparison table of target detection model performance.

|  | YOLO v4 | SSD |
|---|---|---|
| Training time | 17,504 s | 23,581 s |
| Average training time per round | 87.52 s | 117.91 s |
| Number of iterations (times) | 35,400 | 35,200 |
| Average time required per iteration | 0.4945 s | 0.6699 s |
| Batch size | 4 | 4 |
| Allowable maximum batch size | 16 | 4 |
| Convergence speed (round) | 140 | 51 |
| Time required for convergence | 12,362 s | 6029 s |
| Testing time | 229 s | 280 s |

The accuracy and recall of the detection results and the F1 score are important performance indicators of the target detection model. As shown in Table 5, the precision of YOLO v4 model was 0.9934, the recall was 0.7358, and the F1 score was 0.8454, which were higher than those of SSD at 0.9761, 0.6880, and 0.8071, respectively. A more intuitive accuracy comparison is shown in Figure 10. In summary, the training speed and detection speed of YOLO v4 model were faster than those of SSD model, while the precision and recall of YOLO v4 model and F1 scores were also higher than those of the SSD model, so YOLO v4 was more effective than SSD in detecting dead wood of dead nematode-infested pine wood.

**Table 5.** Comparison table of target detection accuracy.

|  | YOLO v4 | SSD |
|---|---|---|
| Manual testing | 1843 | 1843 |
| Model Testing | 1365 | 1299 |
| Correct detection | 1356 | 1268 |
| Precision rate | 0.9934 | 0.9761 |
| Recall Rate | 0.7358 | 0.6880 |
| F1 Score | 0.8454 | 0.8071 |

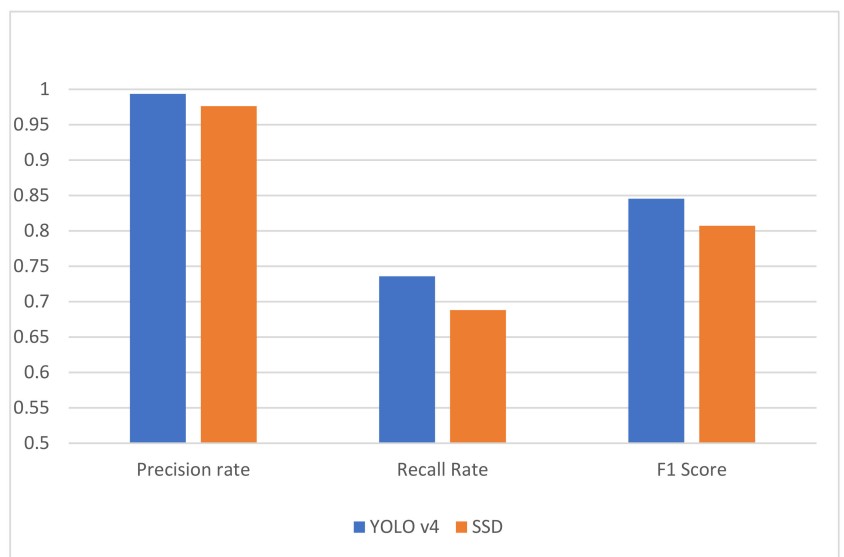

**Figure 10.** YOLO v4 and SSD detection accuracy comparison.

### 3.2. Performance Analysis of Classification Model

A training loss comparison graph (Figure 11) of the recognition classification models reveals that the loss of GoogLeNet decreased the fastest at the start of training and stabilized first. The training loss values of the four other models varied to some extent, and the final training loss of GoogLeNet was the lowest of the four models.

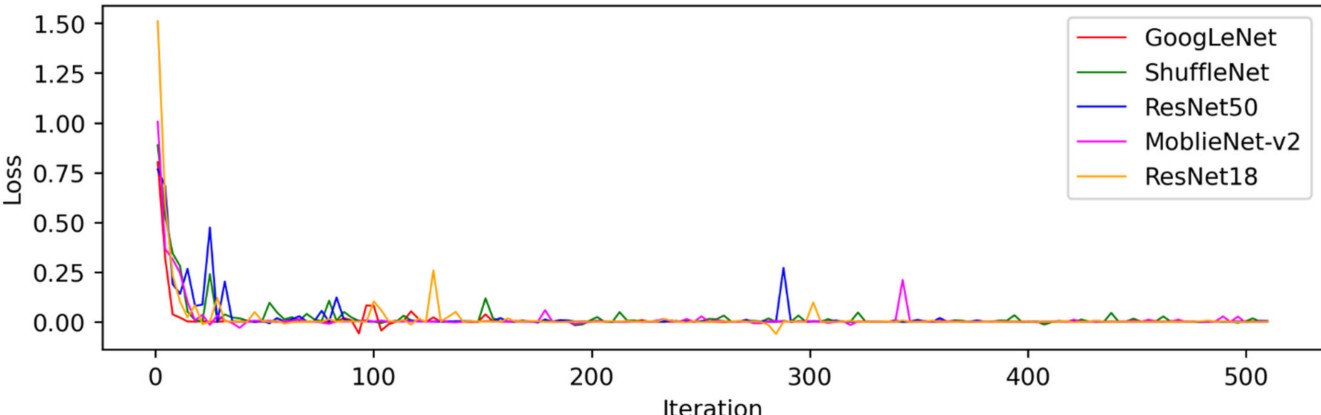

**Figure 11.** Comparison of training loss of recognition classification models.

Analysis of the training accuracy of the recognition classification models (Figure 12) showed that all four models converge with a high classification accuracy level. GoogLeNet's training accuracy increased more steadily and at a faster rate, and reached convergence the quickest, while the training accuracies of the other models slightly increased with iterations.

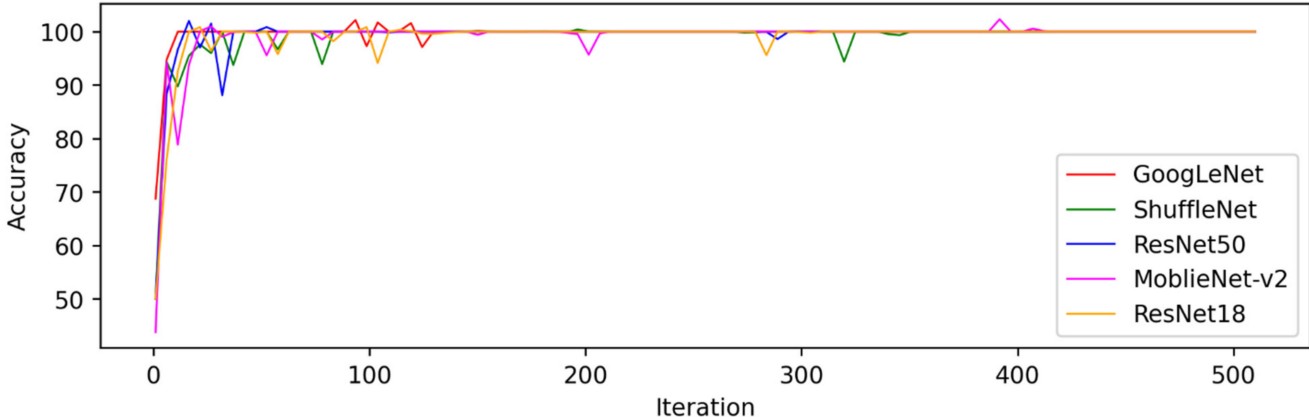

**Figure 12.** Comparison of training accuracy of recognition classification models.

Analyzing the performance comparison table of the recognition classification models, it can be observed in Table 6 that, in terms of the average model training time, GoogLeNet trained significantly faster than the other four models, taking only 79.3 s, while the other four models took more than 90 s, and the final loss value of GoogLeNet was the lowest among all models. For the validation effect, GoogLeNet achieved 100% classification accuracy for the validation set, which was the highest classification accuracy among the four models. Overall, GoogLeNet was the best performer in this recognition categorization.

**Table 6.** Recognition classification model performance comparison table.

|  | GoogLeNet | ResNet50 | MobileNet-v2 | ShuffleNet | ResNet18 |
|---|---|---|---|---|---|
| Training time | 79.3 s | 149.2 s | 156.9 s | 124.9 s | 97.8 s |
| Final loss value | 0.0003 | 0.0019 | 0.0031 | 0.0031 | 0.0010 |
| Validation accuracy | 1 | 0.952 | 0.968 | 0.971 | 0.983 |

### 3.3. Detection Classification Effect Display

To improve the accuracy of precision evaluation, and to provide more effective guidance for pine nematode control, the model was converted into a shapefile of face element type utilizing the conversion connection between picture coordinates and geographic coordinates and projection information. The resulting shapefile stored the target detection confidence information and the discrimination information of dryness degree. Attributes are shown in Figure 13, where Confidence is the target detection confidence information, Class is the discriminant information of dryness, 'ganku' represents dry nematode-infested pine wood, and 'xinku' represents recently dead nematode-infested pine wood.

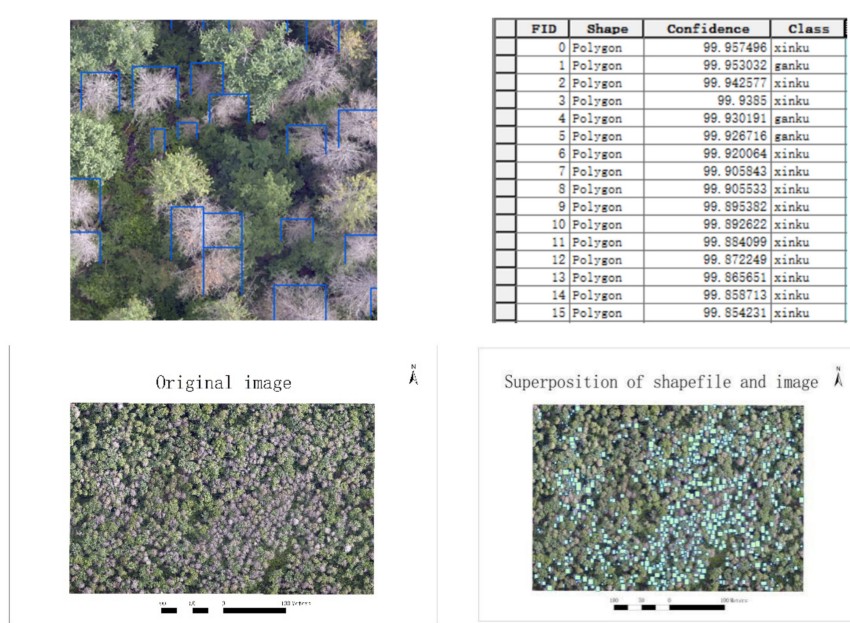

**Figure 13.** (**a**) YOLO v4 final detection effect. (**b**) Shapefile attribute representation intent. (**c**) Original image. (**d**) Shapefile overlaid with image data.

### 3.4. Further Experiments

In the same experimental environment, we choose the YOLO v4 model, which performed better in the experiments, to label the dry and recently dead nematode-infested pine wood in the same dataset separately, forming a dual-target detection dataset that was identical except for the different labeling methods. We conducted a dual-target detection and classification experiment to allow the model to detect and classify dry and recently dead nematode-infested pine wood at the same time, so that the detection and classification tasks were completed simultaneously. We compared and analyzed the experimental results of the dual-target detection and classification method with the staged detection and classification method that used the YOLO v4 model as the first stage target detection model, and the GoogLeNet model as the second stage recognition and classification model. The experimental results were tested with different size sample sets. The accuracy comparison results are shown in Tables 7 and 8, and two more intuitive accuracy comparison graphs are shown in Figures 14 and 15.

**Table 7.** Comparison of detection effect between staged detection classification and dual-target detection classification.

| | Staged Detection Classification Small Sample Set | Dual-Target Detection Classification Small Sample Set | | Staged Detection Classification Large Sample Set | Dual-Target Detection Classification Large Sample Set | |
|---|---|---|---|---|---|---|
| Number of pictures | 603 | 603 | | 884 | 884 | |
| Withering degree of dead nematode-infested pine wood | dead | Recently dead | Dry | dead | Recently dead | Dry |
| Number of labels | 7042 | 5472 | 1570 | 10,145 | 8365 | 1780 |
| Total number of labels | 7042 | 7042 | | 10,145 | 10,145 | |
| Manual detection of dead nematode-infested pine wood | 1843 | 1843 | | 1843 | 1843 | |
| Model detection of dead nematode-infested pine wood | 1035 | 752 | | 1365 | 1388 | |
| Correct detection of dead nematode-infested pine wood | 1016 | 747 | | 1356 | 1369 | |
| Precision rate | 0.9816 | 0.9934 | | 0.9934 | 0.986 | |
| Recall Rate | 0.5513 | 0.4053 | | 0.7358 | 0.7428 | |
| F1 Score | 0.7061 | 0.5757 | | 0.8454 | 0.8473 | |
| Training time | 7729 s | 7813 s | | 17,504 s | 18,262 s | |
| Test time | 218 s | 217 s | | 229 s | 242 s | |

**Table 8.** Comparison of classification effects between classification by staged detection classification by dual-target detection methods.

| | Staged Detection Classification Small Sample Set | | Dual-Target Detection Classification Small Sample Set | | Staged Detection Classification Large Sample Set | | Dual-Target Detection Classification Large Sample Set | |
|---|---|---|---|---|---|---|---|---|
| Withering degree of dead nematode-infested pine wood | Recently dead | Dry | Recently dead | Dry | Recently dead | Dry | Recently dead | Dry |
| Number of labels | 200 | 200 | 5472 | 1570 | 200 | 200 | 8365 | 1780 |
| Total number of labels | 400 | | 7042 | | 400 | | 10,145 | |
| TP | 380 | | 372 | | 337 | | 343 | |
| FN | 5 | | 17 | | 1 | | 7 | |
| TN | 641 | | 356 | | 1013 | | 1034 | |
| FP | 9 | | 7 | | 3 | | 4 | |
| Classification Accuracy | 0.9865 | | 0.9681 | | 0.989 | | 0.9921 | |
| Test time | 218 s | | 217 s | | 229 s | | 242 s | |

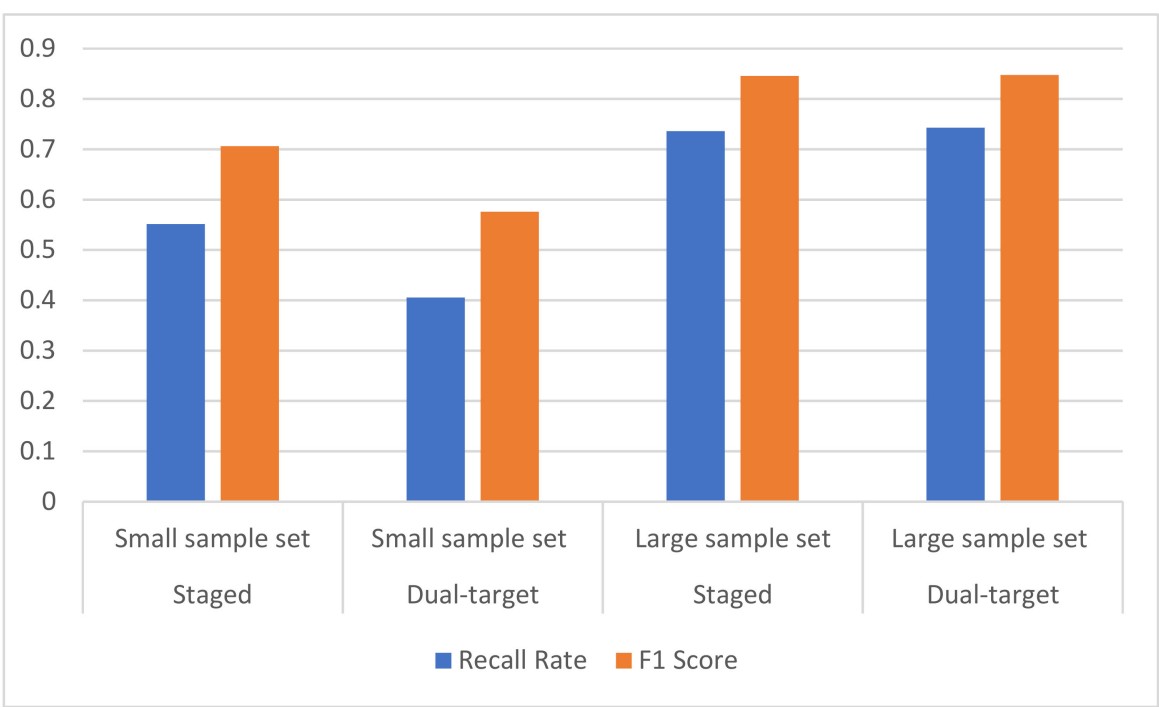

**Figure 14.** Comparison of the detection accuracy of the staged detection classification method and dual-target detection classification method.

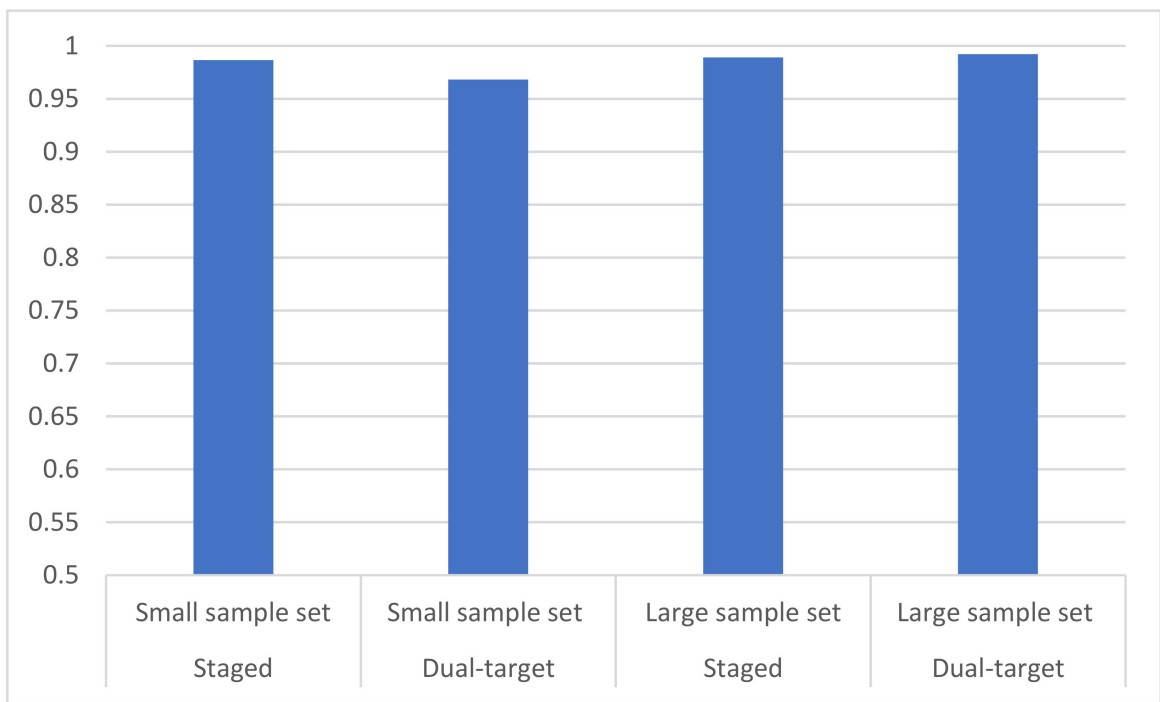

**Figure 15.** Comparison of the classification accuracy of the staged detection classification method and dual-target detection classification method.

Analysis of Table 7 and Figure 14 reveals that the training and detection speeds of the staged and dual-target detection and classification methods were comparable. When using the same small sample set, the precision rates of both methods reached high, similar values, i.e., 0.9816 and 0.9934, respectively. However, the recall rate of the former was 0.5513 and the F1 score was 0.7061, which are better than the recall rate of the latter, 0.4053,

and its F1 score of 0.5757. When employing the same big sample set, the precision rates of both methods were very high, and similar at 0.9934 and 0.9860, respectively. They also performed similarly in terms of recall and F1 score.

The above data show that when using the same small sample set, the staged detection and classification method outperformed the dual-target detection and classification method, while when using the same large sample set, the detection effects of both methods were comparable. In other words, when data volume was limited, the detection effect for dead nematode-infested pine wood of the staged detection and classification method was better than that of the dual-target detection and classification method, and when the data volume was sufficient, the two approaches had similar detection performance.

It can be observed in Table 8 and Figure 15 that both the staged and dual-target detection and classification methods achieved high accuracy for the wilting degree classification of dead nematode-infested pine wood with similar detection times. However, the former used a dataset of only 400 tags, which is much smaller than the dataset of more than 7000 tags used by the latter. In addition, the training time of the staged method was only 79.3 s compared to the dual-target model, which makes it easier to build and adjust the sample dataset and experimental parameters, making the classification part of the experiment more flexible and lightweight.

In summary, when using the same small sample set, the detection effect for dead nematode-infested pine wood of the staged detection and classification method using a combination of the YOLO v4 model and the GoogLeNet model was better than that of the dual-target approach using only the YOLO v4 model, whereas when using the same large sample set, the detection effects were comparable. That is, the combination of GoogLeNet and YOLO v4 can further improve the detection accuracy of YOLO v4 in detecting dead nematode-infested pine wood when data are limited, but this cannot be achieved when the data are large enough. On the other hand, although the staged and the dual-target approaches had similar effects on the wilting degree classification of dead nematode-infested pine wood when using the same sample set, the dataset employed in the former way much smaller than that in the latter, and the recognition classification model used in the staged method was faster to train and more flexible for adjusting and constructing the model. Therefore, while retaining the fast training and detection characteristics of the single-stage target detection model, the staged detection and classification method also reduced the dependence of the target detection model on data volume, further improving the detection accuracy of the model for dead pine nematodes with limited data volume, and was more flexible in achieving accurate wilting degree classification of dead nematode-infested pine wood.

## 4. Discussion

Pine nematode disease has caused significant damage to forests in many countries [2], and detecting pine nematode-infested wood remains a critical challenge for researchers. Traditional detection methods suffer from issues such as heavy workload, low efficiency, strong subjectivity, and limited capacity for large-scale monitoring [39]. To overcome these limitations, deep learning technology can be applied to achieve rapid and automated detection and classification of nematode-infested wood based on high-resolution images.

In this study, we utilized helicopter aerial images of forest areas with a spatial resolution of 0.03 m as the database and employed a combination of YOLOv4 and GoogLeNet models for detecting and classifying dead nematode-infested pine wood in the images. We used the YOLO v4 model to detect dead nematode-infested pine wood in the images, followed by the GoogLeNet model to classify them as dry and recently dead nematode-infested pine wood. The final F1 score, detection precision, recall, and wilt degree classification accuracy were 0.8454, 0.9934, 0.7358, and 0.9890, respectively. Compared with the F1 score of the detection result of the recognition network SCANet proposed by Qin et al. (0.79) [18] and the optimized algorithm of a multi-scale attention-UNet model by Ye et al., with a recall rate of 0.57 [22], the detection results of this study were better.

When the sample size is insufficient, researchers usually use rotation, adjusting brightness, and other methods, to increase the sample size [40,41]. To improve the model's detection and classification ability, we extended the training dataset by performing image flipping, scaling, and color dithering operations. However, it is worth noting that the validation and test sets were not subjected to data enhancement operations to obtain unbiased estimates. Furthermore, we conducted comparative experiments using the staged detection and classification approach and the YOLO v4 method at different scale sample sizes. The results show that the staged detection and classification approach had higher detection accuracy than the method using only the YOLO v4 model when the data volume was limited. We believe that the addition of GoogLeNet improves the YOLO v4 model, by allocating more image features extracted from a limited dataset to the target detection task, resulting in higher detection accuracy. This is important in practical applications, where data availability can be a limiting factor. The proposed method is also more flexible in achieving accurate classification tasks, which has implications for preventing and controlling pine nematode disease epidemics in forest areas.

Despite the promising results, some limitations were also found. The proposed method was tested on a limited dataset and more research is needed to validate its performance on larger and more diverse datasets, including lower resolution datasets. In addition, the images used in the current study included only red, green, and blue bands, and it is necessary to include supplementary data, such as near-infrared band data, for additional experiments and comparative analysis. There are some relevant studies on in this regard, e.g., [7,9]. Hardware limitations also limited the models tested in the experiments, and incorporating cloud computing may help to improve hardware conditions and expand the types of models tested.

We encountered challenges in detecting low-height pine woods that were hidden in the shadows of surrounding taller trees, which led to missed detections and reduced recall rate. Furthermore, healthy trees were sometimes incorrectly identified as dead nematode-infested pine wood, resulting in reduced precision rate. To address these issues, we plan to utilize point cloud data that contains height information as auxiliary data and improve the network structure of the current target detection model to enhance the recall and precision of detecting dead nematode-infested pine wood. There is other, current research to draw on in this regard, e.g., [36,37,42].

In conclusion, the presented staged detection and classification approach has great promise for detecting and classifying dead nematode-infested pine wood, and has the potential to contribute to more effective and efficient pine nematode disease epidemic prevention and control in forest areas.

## 5. Conclusions

A staged detection and classification method is proposed to combine the YOLO v4 model and GoogLeNet model for target detection of dead nematode-infested pine wood in the images as well as wilting degree classification, with high spatial resolution, based on data from helicopter-acquired images of forest areas. The main findings are as follows.

1. In terms of dead nematode-infested pine wood target detection, the YOLO v4 model outperformed the SSD model with a detection accuracy of 0.9934, a recall of 0.7358, and an F1 score of 0.8454.
2. For the wilting degree classification of dead nematode-infested pine wood, the GoogLeNet model outperformed the remaining four models, namely ShuffleNet, ResNet50, MobileNet-v2 and ResNet18, with a classification accuracy of 0.9890.
3. In the case of limited data volume, the staged detection and classification method combining GoogLeNet model and YOLO v4 model improved the detection accuracy of YOLO v4 model for dead nematode-infested pine wood while retaining the fast training speed and detection speed of YOLO v4 model, and achieved the accurate classification of wilt degree of dead nematode-infested pine wood in a more flexible way. With a sufficient amount of data, the detection accuracy of both was comparable.

**Author Contributions:** Conceptualization, X.Z. and R.W.; methodology, X.Z.; software, X.Z.; validation, X.Z., Q.Y., W.S. and X.L.; writing—original draft preparation, X.Z.; writing—review and editing, X.Z.; visualization, X.Z.; supervision, X.Z. and X.C.; project administration, R.W.; funding acquisition, R.W. All authors have read and agreed to the published version of the manuscript.

**Funding:** National Natural Science Foundation of China 'biomass precision estimation model research for large-scale region based on multi-view heterogeneous stereographic image pair of forest' (41971376).

**Institutional Review Board Statement:** Not applicable.

**Informed Consent Statement:** Not applicable.

**Data Availability Statement:** Not applicable.

**Conflicts of Interest:** The authors declare no conflict of interest.

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
