# Peer review of "Automatic Detection and Classification of Dead Nematode-Infested Pine Wood in Stages Based on YOLO v4 and GoogLeNet"

_forests, doi:10.3390/f14030601_

Round 1

Reviewer 1 Report

Finding and clearing infected pine trees in forests helps decrease pine wood nematode disease. Deep learning's target detection model has been used to monitor pine nematode-infested wood automatically using YOLO v4 and GoogLeNet techniques in this paper titled “Automatic detection and classification of dead pine nematode wood in stages based on YOLO v4 and GoogLeNet”. The authors also established comparisons between these TWO techniques.

I would say the topic is relevant and needed to push the current body of knowledge in this field. Therefore, the paper is relevant to be published.

This article provides comparison between YOLO v4 and GoogLeNet techniques that is interesting. In the recent past, people have published papers on YOLO v5 and multi-band feature fusion transformer. Similarly, some scientists have used other deep-learning models to detect or predict pine wood nematode disease.

The methodology seems fine. However, there are serious issues with references and in-text citations.

There are serious issues with in-text citations whose references are provided in the reference list. Some of the papers I could not find while googling. It means some of the references in the list are made-up and pose research ethics question for authors to get them corrected. For example, check out in-text citations for reference 15 and 33. You should carefully check all of the given references in your paper and cite them properly those papers whose reference one can find easily on google scholar.

General Comments

The study and manuscript is fine except few critical revisions to be made. There are some serious issues in the references. The random check of references has returned with no hits on GOOGLE SCHOLAR e.g., references 15 and 33 and some more. The authors need to recheck the citations and references again carefully.

Author Response

   Thanks for your very careful review and very valuable suggestions. Your reviews have substantially improved the quality of our manuscript. For the reply to you, please see the attachment.

Reviewer 2 Report

The manuscript on the detection of pine wood nematode infestation using YOLO and GoogLeNet is very interesting and provides a very useful tool that, when applied, it can substantially assist the expansion of infestation. The methods are adequately described and the authors present everything in details. There are however specific points that need to be improved and revised before the manuscript can be accepted for publication in the journal. Nevertheless, I firmly believe that once these points are addressed the manuscript can be accepted. 

1. Remove the last two sentences of the abstract - you do not need to go into so much arithmetic details in an abstract. 

2. In the manuscript you use the term "Dry pine nematode wood" and "New dead pine nematode wood". I believe that these terms should be replaced by "Dry nematode-infested pine wood" and "Recently dead nematode-infested pine wood" as these would be more accurate. Please check that throughout the text. 

3. In Lines 280-290, there is a mix up in the terms and symbols used in the formulas. For example TP is given two different explanation (Line 281 and Line 291). Please have a look at it and correct accordingly so that the reader can easily follow the text and use of formulas. 

4. Are the differences between YOLOv4 and SSD statistically significant? Is there a way to compare the values you provide in Tables? The same question goes to the other values you present in Tables 5 and 6. 

5. Discussion is relatively weak, as the authors simply present the results. This part should be more elaborated and enhanced with references that would give the reader the opportunity to relate these results with previous studies.

Author Response

(The authors gave the same response as above.)
